# Infections, Vaccines and Autoimmunity: A Multiple Sclerosis Perspective

**DOI:** 10.3390/vaccines8010050

**Published:** 2020-01-28

**Authors:** Dejan Jakimovski, Bianca Weinstock-Guttman, Murali Ramanathan, Michael G. Dwyer, Robert Zivadinov

**Affiliations:** 1Buffalo Neuroimaging Analysis Center, Department of Neurology, University at Buffalo, State University of New York, Buffalo, NY 14203, USA; 2Jacobs MS Center, Department of Neurology, University at Buffalo, State University of New York, Buffalo, NY 14203, USA; 3School of Pharmaceutical Sciences, University at Buffalo, State University of New York, Buffalo, NY 14214, USA; 4Center for Biomedical Imaging at Clinical Translational Science Institute, University at Buffalo, State University of New York, Buffalo, NY 14203, USA

**Keywords:** multiple sclerosis, immunization, DNA-vaccine, T-cell vaccine, influenza, hepatitis B, Epstein Barr virus

## Abstract

**Background:** Multiple sclerosis (MS) is a chronic neuroinflammatory and neurodegenerative disease that is associated with multiple environmental factors. Among suspected susceptibility events, studies have questioned the potential role of overt viral and bacterial infections, including the Epstein Bar virus (EBV) and human endogenous retroviruses (HERV). Furthermore, the fast development of immunomodulatory therapies further questions the efficacy of the standard immunization policies in MS patients. **Topics reviewed:** This narrative review will discuss the potential interplay between viral and bacterial infections and their treatment on MS susceptibility and disease progression. In addition, the review specifically discusses the interactions between MS pathophysiology and vaccination for hepatitis B, influenza, human papillomavirus, diphtheria, pertussis, and tetanus (DTP), and Bacillus Calmette-Guerin (BCG). Data regarding potential interaction between MS disease modifying treatment (DMT) and vaccine effectiveness is also reviewed. Moreover, HERV-targeted therapies such as GNbAC1 (temelimab), EBV-based vaccines for treatment of MS, and the current state regarding the development of T-cell and DNA vaccination are discussed. Lastly, a reviewing commentary on the recent 2019 American Academy of Neurology (AAN) practice recommendations regarding immunization and vaccine-preventable infections in the settings of MS is provided. **Conclusion:** There is currently no sufficient evidence to support associations between standard vaccination policies and increased risk of MS. MS patients treated with immunomodulatory therapies may have a lower benefit from viral and bacterial vaccination. Despite their historical underperformance, new efforts in creating MS-based vaccines are currently ongoing. MS vaccination programs follow the set back and slow recovery which is widely seen in other fields of medicine.

## 1. Introduction

Multiple Sclerosis (MS) is a chronic inflammatory and neurodegenerative disease of the central nervous system (CNS) that significantly contributes towards neurological disability of the young and working population [1]. Although the etiology of MS still remains largely unknown, multiple environmental risk factors have been associated with greater disease susceptibility and disease progression [2]. In the mid-20th century, associations between the onset of several viral, bacterial, and helminthic infections with the prevalence of MS contributed towards creation of the “hygiene hypothesis” [3]. As with atopy, asthma, and other autoimmune diseases, MS can be partially described through the immunological framework of a balance between the type 1 immune response [T-helper 1 (Th1), driven by viral and bacterial infections that are highly active in autoimmune diseases) and type 2 immune response (Th2, driven by helminth infection and allergic diseases) [4]. As such, the hygiene hypothesis has been initially depicted by a virtually dichotomous relationship between high prevalence of *Trichuris trichiura* (common human helminth as surrogate marker for infections and community sanitation) and low prevalence of MS [5]. As additional support, greater exposure among siblings is also associated with a lower risk of MS, reinforcing a hygiene-based relationship which is already corroborated as a factor in atopy and asthma [6]. Therefore, it can be contemplated that insufficient immune activation early in life would lead to a less-developed and abnormal regulatory network, which can later allow aberrant immune responses towards self-antigens. On the other hand, a higher number of bacterial and viral infections during childhood that are accompanied with frequent use of antibiotics (all contributing towards Th1 response) are associated with greater risk of MS [7]. More so, the road towards the discovery of the first MS disease modifying treatment (DMT) – interferon-β - was mainly driven by a viral hypothesis in mind [8]. That being said, most current MS DMTs (either small molecule or antibody-based medications) are focused on selective or general suppression of the immune system. MS-based medications would either attempt at depleting entire subset of immune cells (selective B-cell or both B and T-cell depletetors like ocrelizumab and alemtuzumab, respectively), sequester the pathological immune cells away from the CNS (natalizumab and sphingosine-phosphate receptor modulators), or inhibit the expansion of stimulated lymphocytes (teriflunomide and cladribine). [9,10]. However, none of these interventions target the underlying pathophysiology that still remains largely unknown.

Due to ongoing concerns regarding the role of viruses, bacteria, and their specific vaccinations in MS patients, the aim of this narrative review is to summarize such findings. Furthermore, we will discuss the attempts at developing vaccine-based therapies that would counter the immune imbalance present in MS. Lastly, we outline the potential effect of current MS immunomodulatory treatment on vaccination efficacy and comment on the latest 2019 American Academy of Neurology (AAN) immunization recommendations.

## 2. Infections and Multiple Sclerosis

### 2.1. Bacterial Infections

Recent studies have described potential associations between the rate of certain bacterial infections and prevalence of autoimmune or neurological diseases such as MS and Alzheimer’s disease, respectively [11]. The mediators through which these factors interact include: changes in gut microbiota, Westernization of lifestyles, and improvements in sanitation. *Helicobacter pylori* is a highly motile Gram-negative bacteria that infects as much as 50% of the world’s population and contributes towards gastric ulcers, chronic gastritis, and gastrointestinal cancers. Apart from the known associations between successful treatment of *H. pylori* and improvement in idiopathic thrombocytopenic purpura, studies have suggested that *H. pylori* infection can potentially influence the risk of MS [11,12]. The initial report of a potential protective association between higher rates of *H. pylori* infection in MS patients came from a Japanese study which analyzed 162 patients with either neuromyelitic optica (NMO) or MS [13]. The study showed that *H. pylori* seropositivity was significantly lower in MS patients when compared to the rate seen in NMO patients [13]. Furthermore, not only do MS patients have lower rates of *H. pylori* infection rates when compared to healthy controls, but *H. pylori* seropostivie MS patients also have lower disability scores [14]. Two recent meta-analyses including more than 1500 MS cases and matched healthy controls have confirmed the findings of lower prevalence of *H. pylori* in MS patients [15,16]. Experimental MS mice studies have also shown that *H. pylori* infection contributes towards lower disease severity and significant reduction in the number of CD4^+^ T-cells [17]. Interestingly, data have also shown that presence of a single filamentous bacterium in the small intestine can contribute to significant T-cell activation and induction of mural pro-inflammatory Th17 response targeting other pathogenic colonies [18]. Evolutionary, these symbiotic processes would maintain the balance of the gut bacterial colonies; however, the higher Th17 cytokine levels may also contribute to greater rate of systemic autoimmune diseases [18]. Additionally, studies describing the inverse association between a lower *H. pylori* infection rate and high prevalence of allergic diseases have also pointed towards a *H. pylori*-derived VacA toxin [19]. This toxin is responsible for shifting the balance towards Th1-based cytokine release, induced by release of IL-18 and conversion of CD4^+^ to regulatory CD25^+^Foxp3^+^ T-cells [19].

On the other hand, several reports have shown a greater presence of antibodies towards *Chlamydia pneumoniae* in the cerebrospinal fluid (CSF) of MS patients when compared to controls [20]. Analysis of more than 120,000 members of California-based neurological practice has shown that presence of *C. pneumoniae* antibodies are significantly associated with a 70% increase in future MS diagnosis; however, these findings are highly variable in the literature, with negative findings seen in other cohorts [21]. Later studies have also shown that *C. pneumoniae* can be seen in a vast range of neurological diseases [22]. Therefore, the high prevalence of *C. pneumoniae* can be associated with either some pathophysiological relevance or only a consequence of the general inability to fight-off such pathogens [22]. The induction of Th1-based inflammation due to bacterial infections can be highlighted by the exacerbation of MS symptoms due to acute urinary and/or respiratory tract infections [23]. Lastly, the effect of infections on the risk of MS can potentially be assessed through the proxy of antibiotic use. Although use of antibiotics specifically targeting *C.pneumoniae* do not influence the MS risk, studies have shown that overall use of antibiotics is associated with lower risk of MS [24,25]. However the trend of opposing findings continues, with other studies showing that the pairing of higher bacterial infection rates with greater antibiotic use are associated with increased risk of future MS diagnosis [7].

### 2.2. Viral Infections and Virus-Targeted Treatments in Multiple Sclerosis

Epstein-Barr virus (EBV) is a γ-herpesvirus and causative agent for infectious mononucleosis (IM) as well as lymphoid and epithelial malignancies. Furthermore, both asymptomatic EBV infection and IM have been also associated with an increase in MS susceptibility [26,27]. Despite the high EBV seropositivity of the general population (~90%), MS patients exhibit virtually omnipresent EBV reactivity [28]. In an analysis of 1,047 clinically isolated syndrome (CIS) patients, only 1 (<0.01%) patient was deemed to be truly EBV seronegative [28]. The imminent increase in MS risk following an EBV seroconversion has been elegantly shown by a study which utilized serial blood samples derived from more than 8 million active duty military personnel [29]. During the follow-up period of 12 years, 315 soldiers were diagnosed with MS and later compared to randomly selected matched controls [29]. Ten (10) of these MS patients (3.3%) were initially EBV seronegative and all of them had a change in serostatus prior to diagnosis [29]. The average time from the first seropositive test to MS diagnosis was 3.8 years [29]. On the other hand, only 35.7% of the seronegative healthy controls exhibited EBV seroconversion [29]. Similar findings were seen in pediatric-onset MS cases, where EBV seropositivity was the only serological test which significantly differentiated children with MS and controls [30]. In that study, 86% of children with MS had evidence of previous EBV infection when compared to only 64% in the matched control group [30]. The findings of greater risk for MS in people with a high titer of EBV antibody levels were also corroborated, regardless of the ethnic/racial characteristics of the study population [31]. Interestingly, the study associations of lower MS risk in Hispanics born in low- or medium-income countries was deemed as non-significant after correcting for the EBV status [31]. Lastly, a recent Finnish maternity cohort of more than 800,000 women showed that high levels of EBV viral capsid (VCA) antibodies during the pregnancy period is associated with greater MS risk in their offspring [32]. This study further corroborated the role of EBV showing that mothers with high levels of Epstein-Barr nuclear antigen 1 (EBNA-1) antibody titer had more than 3 times greater risk of future MS development [32]. In addition to increasing MS risk, higher EBV load as measured by the serum-based antibody titer has been associated with greater clinical and radiological MS pathology [33,34]. For example, a spike in immunological response towards EBV early antigens and towards EBV DNA is associated with concurrent disease activation, resulting in an MS relapse [33]. Moreover, higher anti-EBNA-1 levels are also associated with greater irreversible disability progression over mid-term follow-up period [35]. Lastly, the clinical outcomes coincide with associations of higher anti-EBNA-1 levels and greater MRI-derived rates of brain atrophy, more focal MS lesions, and more extensive cerebral demyelination [36,37].

Multiple experimental and histopathological studies have proposed potential mechanisms which can explain the interactions between EBV and the immunopathophysiolocal MS cascade. EBV infection significantly activates B-cells and upregulates their ability to present antigens [38]. Now as potent antigen-presenting cells, the infected B-cells will more efficiently process the relevant myelin autoantigen, attach it to their major histocompatibility complex II (MHC-II) receptors and cross-present them to the pathogenic CD8^+^ T-cells [38,39]. In order to ensure survival, EBV also immortalizes the host B-cells, which in turn can continue to activate the effector T-cells. Although EBNA-1 molecule is not essential in the viral survival within the host B-cells, the presence of EBNA-1 induces up to a 100 times faster B-cell immortalization rate [40]. Lastly, EBV-infected B-cells employ multiple collateral activation pathways that can activate the aforementioned T-cells and enhance their IL-17 production [41]. High numbers of EBV-infected B-cells are repeatedly observed in MS white matter (WM) lesions, cortical lesions, and surrounding meninges [42,43]. The prevalent intracerebral EBV presence has been also demonstrated in a recent study where up to 93% of examined MS brain samples exhibit expression of EBV latent membrane protein 1 (LMP-1) and EBV-encoded RNA [44]. In terms of molecular mimicry, a recent study has shown partial complementary sequences and antibody cross-reactivity between EBNA-1 and newly proposed MS antigen of anoctamin 2 (CNS-based ion channel) [45]. A selected group of MS patients (approximately 32%) do present with increased reactivity towards Anoctamin 2 and may represent an EBV-induced MS sub-phenotype [46]. All aforementioned findings provide multiple pathways that will aim at determining and isolating molecular targets in a potential vaccine-based EBV-targeted MS treatment.

Due to the complex nature of human herpesviruses, their life cycle, multitude of antigens being expressed, and the known oncogenic effects, developing a live vaccine towards EBV is a highly challenging proposition. Apart from a successful vaccination for α-herpesvirus Varicella-Zoster virus (VZV) and vaccination for the veterinary and highly oncogenic Marek’s disease virus (MDV), the use of live-attenuated oncogenic herpesviruses remains debated [47,48]. Therefore, multiple alternative vaccine candidates include targeting EBV-based glycoproteins (gp), EBV lytic proteins, and EBV latent proteins (summarized in Table 1) [49]. Furthermore, the advances in nanotechnology regarding assembling nanoparticles, scaffolds, and microneedles can allow better antigen delivery and increase the selected antigen circulation time [50].

At this time, the biggest success towards developing an EBV-based vaccine has been by targeting gp350. This type I glycoprotein is crucial for the ability of EBV to enter the host B-cells by binding their CD21 or CD35 receptor. Both phase I and II human trials of gp350 vaccines have been carried out [51]. In a first phase I/II trial, 81 EBV seronegative healthy volunteers were vaccinated with 0.5 ml recombinant gp350 at baseline, one month later, and six month booster doses [52]. A month after the last dose, all participants had significant anti-gp350 antibodies and no significant adverse events were reported. Although at a lower response rate, the EBV neutralization was most significant in the group vaccinated with adjuvanted gp350-specific vaccine (60.9% of study participants achieved responses over the desired cut-off value) [52]. The same group also conducted a follow-up phase II double-blind, randomized, placebo-controlled (1:1 ratio) trial which enrolled 181 EBV seronegative participants [53]. After two years of follow-up, 8 out of 91 placebo-controlled participants developed IM and only 2 out of 90 gp350-vaccinated developed mild cases of IM (78% reduction in IM occurrence) [53]. It is important to note that the two vaccinated subjects developed the disease within their first six months within the trial and before the full three dose vaccination protocol was completed. However, the vaccination did not have any significant effect on the rate of asymptomatic EBV infection [53]. Therefore, one would assume that such vaccination with gp350 may result only in a decrease in the total EBV load and prevent progression into IM manifestation [53]. Given the vast variability and the long prodromal period from an EBV infection to the onset of MS, such EBV vaccine trials in MS populations are not feasible. However, until the successful development of a potent EBV vaccine, an alternative empirical study of EBV vaccination in first degree MS relatives can be hypothesized. With a 30 times greater rate of MS occurrence in first degree relatives when compared to unrelated populations, such intervention may potentially decrease the overall MS incidence [54].

Other potential targets for vaccine development include immediate and early EBV proteins that are expressed before the formation of the viral evasion capabilities. Therefore, both Zta and Rta immediate proteins (encoded by *BZLF1* and *BRLF1*, respectively) are easily recognizable due to a uninhibited CD8^+^ T-cell response [55]. Furthermore, the early lytic proteins BMLF1 and BMRF1 can be detected by CD4^+^ T-cells as early as the first day of EBV infection [56]. Studies have examined the utility of BZLF1 vaccine in mice models of EBV-induced post-transplant lymphoproliferative disorder and showed successful induction T-cell immunity towards the infected tumor cells [57]. Lastly, recent evidence also show that the aforementioned latent proteins (EBNA) can be recognized by CD8^+^ and CD4^+^ T-cells and prevent further expansion of EBV-infected B-cells [56]. In the wake of understanding the importance of B-cells in the MS pathophysiology, this type of vaccine intervention would potentially exert a therapeutic outcome [58].

Another potential EBV-targeted therapy could utilize EBV-primed CD8^+^ T-cells. Studies have shown that T-cells directed towards EBV antigens become increasingly exhausted, resulting in a gradual decrease of available EBV-specific CD8^+^ response [59]. This finding is further supported by associations of greater EBV replication rate, higher anti-EBNA-1 activity, and lower EBV-specific CD8^+^ cell count [59]. Mice studies have demonstrated that adoptive transfer of latent EBV antigen-specific CD8^+^ T-cells (either towards BMLF1 or LMP2) can reduce the blood viremia and target lyrically-replicating EBV-transformed B-cells [60]. Furthermore, a histopathological study showed significant infiltration of MS lesions with cytotoxic CD8^+^ T-cells that can recognize latent and lytic EBV proteins [61,62]. In spite of concerns regarding the risk of allogenic viral-specific T-cells to recognize recipient HLA molecules, the use of adaptive transfer has generally not resulted in graft-versus-host disease [63]. An open-label phase I trial enrolled 10 progressive MS patients and infused them with escalating doses of in-vitro primed EBV-specific autologous T-cells [64]. The antigen-specific target in these T-cells was towards the aforementioned EBNA-1, LMP1, and LMP2A [64]. When compared to patients receiving low-activity T-cells, all six patients that received highly EBV-reactive T-cells exhibited some level of clinical improvement (3/6 patients exhibited decrease in disability scores) [64]. The lack of improvement in the remaining patients was attributed to substantial irreversible pathology at the time of enrollment [baseline Expanded Disability Status Scale (EDSS) scores of 8.0] [64]. However, it is unknown whether the presence of cerebral EBV-specific CD8^+^ T-cells are contributing towards the local inflammatory processes and damaging the surrounding neuronal tissue or if they are part of a protective anti-viral mechanism. Before designing large-scale human trials, a greater understanding of the role of such EBV-targeting T-cells is warranted.

The increased expression of MS-associated retrovirus (MSRV) has been suggested as a potential link that would connect the EBV infection and its effect on the MS pathophysiological processes [65]. This virus belongs to the human endogenous retrovirus (HERV) family (later also interchangeably termed as HERV-W) which entered the mammalian germ line millions years ago via transmission through both Mendelian and non-Mendelian processes, and contributes towards 8% of the total human DNA [66]. Based on the specific tRNA involved in the HERV translation, these viruses are classified in several subfamilies including HERV-H, HERV-K, and HERV-W and all of them have been previously implicated in MS patients [67,68]. As such, increased expression levels of HERV-W viral proteins (either *env*, *pol* or *gag*) have been detected in MS-derived blood cells, the CSF, and brain tissues [68]. Up to 80% of MS patients present with significant serum levels of HERV-W *env* protein levels, compared to no such findings within healthy controls and very low prevalence within other neurological diseases [69]. Histopathological analysis also demonstrated significant HERV-W *env* immunoreactivity within active MS lesions, collocating HERV-W *env* to macrophages within the perivascular space [69]. Recent meta-analysis corroborated the findings by showing a strong association between MS and MSRV/HERV-W *pol* and *env* expression [68]. This was accompanied by a high odds ratio of 22.7 with no study inconsistencies and publication biases [68]. As a response towards the accumulating HERV-W pathophysiological data, development of antibodies that would target the *env* protein are currently ongoing [70]. GNbAC1 (later renamed as temelimab) is humanized IgG4 monoclonal antibody that binds the entire HERV-W *env* and inhibits the surface domain region, which has the ability to bind the Toll-like receptor 4 and induce pro-inflammatory cytokine release [71]. Apart from the desired target site, temelimab exhibits only one more off-target binding affinity towards MSRV-related protein encoded by chromosome 7 and expressed in the human placenta termed enverin [70]. Encouraged by the good safety profile and linear pharmacokinetics seen in phase I, this antibody has been further tested in two phase IIa and IIb MS trials [72]. In addition to a decrease in MSRV*env* and MSRV*pol* transcripts levels, the group on the highest dose of 18mg/kg GNbAC1demonstrated 2-year significant 63% reduction in new T1-black holes, significant reduction of brain atrophy ranging from 72% reduction in the thalamus to 29% in the whole brain volume (CHANGE-MS; NCT02782858). Interestingly, the study did not show any effect on the number of gadolinium-enhancing lesions. These neuroprotective findings were further corroborated in a 96 weeks extension trial (ANGEL-MS; NCT03239860), which demonstrated sustained whole brain atrophy reduction effects of ~40% in the 18 mg/kg GNbAC1 study group.

Another potential mechanism by which GNbAC1 can exert its beneficial effect is by uncoupling the process of MSRV*env*-mediated reduction in oligodendrocyte differentiation [73]. The presence of MSRV*env* proteins in proximity of oligodendrocyte progenitor cells result in increased cell stress response, formation of 3-nitrotyrosine, and lower myelin basic protein expression [73]. Potentially remyelinating clinical effects can also be seen within the phase IIb extension trial, where the 18mg/kg GNbAC1study group demonstrated improvement within the magnetization transfer ratio signal of the normal-appearing WM (MRI-based marker for myelination). In conclusion, the effectiveness of anti-MSRV*env* antibody strengthens the potential role of HERVs in MS.

## 3. Vaccines and Multiple Sclerosis

### 3.1. Vaccines and Risk of MS Onset and MS Relapses

The core of any post-marketing vaccine safety surveillance includes monitoring for adverse events following immunization (AEFI). Events that occur at a greater rate than the normally expected background prevalence raise new concerns and trigger additional investigations. Classic neurological examples of such vaccine safety analysis involve Guillain-Barre occurrence after pandemic influenza immunizations [74]. The interest of a vaccine-induced MS onset spiked during a large French hepatitis B vaccination campaign which occurred from 1995 through 1997. Multiple reported MS cases developing weeks following hepatitis B vaccination, together with some preliminary positive analyses, resulted in a complete suspension of the country-wide vaccination program [75]. In response, multiple international efforts followed to refute the proposed association [76,77]. A US-based nested case-control study was not able to find evidence of increased risk of MS and hepatitis B vaccination [76]. A study from the Canadian British Columbia also investigated the rate of adolescent onset of MS or other demyelinating diseases before and after the initiation of hepatitis B vaccination and found no sufficient evidence for establishing such causative link [77]. Furthermore, a French-based study revisited the rate of CNS demyelination in children vaccinated during and after the questionable time period (1994-2003) and showed no significant increase in disease onset within the pre-established risk period of three years post-vaccination [78]. However, when the study restricted the analysis to include only subjects that were fully compliant with the vaccination guidelines, exposure to the particular Engerix B vaccine was associated with a 2.4-fold greater odds ratio for future confirmed MS diagnosis [78]. The same study group also attempted to determine whether hepatitis B or tetanus vaccination increased the rate of a second relapse in children with a previously-diagnosed demyelinating CNS disease [79]. Although it concluded that hepatitis B and tetanus vaccination do not increase the relapse rate, the study was highly underpowered, with only 33 pediatric MS patients receiving hepatitis B vaccine, of which only six of them actually relapsed within the three year period [79]. Similar analyses expanded the study scope to include influenza vaccination and also showed no significant increase in relapses within the last two months post-vaccination [80]. A recent meta-regression analysis of all studies pertaining to the risk of MS development after hepatitis B vaccination has corroborated the lack of such an association [81].

This topic was recently revisited by a nested case-control study which analyzed the risk of an acquired CNS demyelinating disease occurring within three years of hepatitis B or human papillomavirus (HPV) vaccination [82]. Although the study refuted such associations, vaccination within 30 days was correlated with a 2.3-fold greater occurrence of CNS demyelinating disease, particularly seen within younger populations (<50 years old) [82]. Among cases that developed CNS demyelinating disease, the most implicated vaccines were influenza and diphtheria, tetanus and pertussis (DTP) [82]. Due to the fact that 3 out of the 24 cases had either a family history of MS, previous radiologically isolated syndrome (RIS) diagnosis, or other systemic autoimmune diseases, the authors concluded that the vaccination only resulted in acceleration towards symptomatic demyelination in preexisting disease circumstances [82]. Given the natural history progression in such circumstances, the phenotypical presentation and follow-up relapse could have been potentially anticipated. The risk of MS and other demyelinating disease after HPV vaccination was further investigated in two large Danish and Swedish registries which encompassed almost four million females and two million doses of quadrivalent HPV vaccine spanning over a follow-up period of seven years [83]. Comparison between the unvaccinated and 2-year post-vaccination periods showed no increased risk of MS after the HPV vaccination [83]. Furthermore, administrative claims data from the largest US medical insurance subsidiary showed no increased risk of optic neuritis development within 60 days after HPV vaccination [84]. Recent meta-analysis included six different HPV studies and was not able to determine any increase in the risk for MS [85]. Smaller numbers of studies have examined the risk of MS after diphtheria, tetanus, and pertussis vaccination and have not found any associations [86,87,88].

Despite some reports suggesting an increased rate of exacerbations following an influenza immunization [89], a larger body of evidence show that such associations do not exist [80,90]. For example, a double-blind, placebo-controlled study randomized 104 MS patients to either standard influenza immunization or placebo injection and demonstrated no increased relapse rate within the 6-month follow-up period [91]. Furthermore, when compared to the placebo, the time to relapse was significantly longer in the immunized MS patients [91]. More importantly, vaccinating MS patients with seasonal influenza vaccine can further limit the deleterious effects derived from fully manifesting flu. Experimental studies have shown that active influenza infection can trigger glial activation, increased T-cell, and neutrophil cerebral trafficking, and contribute towards more severe MS exacerbations [92].

In contrast, some studies have discussed a potential protective effect of Bacille Calmette-Guerin (BCG) immunization on the risk and severity of MS [93,94]. In a small MS pilot trial, MRI-derived lesion activity was determined before and after 12 MS patients were injected with a single dose of a BCG vaccine [93]. When compared to the study run-in period, the vaccination resulted with 51% reduction in contrast-enhancing and 57% reduction in active T2 lesions, respectively [93]. A consecutive double-blind trial utilized 73 CIS patients which were randomized to either BCG vaccination group or placebo and were followed by monthly MRIs for six months. In a second, preplanned extension period, all MS patients continued with IFN-β treatment [94]. During the first 6-month period, BCG vaccinated patients had significantly lower amounts of contrast-enhancing lesions (45.9% relative risk reduction), less new enlarging T2 lesions (63.6% relative risk reduction), and less new T1-based black holes (85.1% relative risk reduction) [94]. Moreover, the reduced amount of MRI activity remained low in the BCG+DMT extension study arm [94]. When compared to placebo+DMT, the BCG+DMT group had a significantly lower risk of fulfilling the clinically definite MS criteria during the 5-years follow-up [94]. A similarly designed study is currently investigating the effect of BCG vaccination in RIS patients. (NCT03888924) An interesting experimental study has recently showed significant differences in mycobacterial responses between MS and NMO patients [95]. The MS patients showed a significantly greater immune response towards *mycobacterium avium* and a significantly low rate of response towards the *mycobacterium bovis* (comparable to BCG), findings which were the opposite of those in NMO [95]. These differences may be explained by different types of transmission (*M. avium* through gut vs. *M. bovis* through skin) and the higher probability of *M. avium* to produce an antigen-specific humoral response [96]. In fact, *M. avium* infection in mice MS models results in early disease onset and more severe disease progression [97]. It is currently unknown through which processes BCG vaccination may interact and alleviate the MS pathophysiology.

### 3.2. Vaccine Effectiveness in Multiple Sclerosis

Due to the increasing use of immunosuppressive medications as standard MS care, considerations regarding the effectiveness of vaccination protocols would naturally arise. A variation of MS case-controlled or observational studies have examined the effectiveness of influenza vaccine and/or vaccination towards capsular bacteria. However, before reviewing the effect of the immunomodulatory treatments, it is important to mention that only a few studies have shown a sufficient vaccination effect in untreated MS patients [98]. The Rebif-Influenza Vaccine Study was one of the first designed trials that would investigate the effect of IFN-β use on vaccinations [99]. The primary end point was achieved if the hemagglutination inhibition titers of the vaccinated patients reached ≥40, indicating seroprotection [99]. Out of 163 MS patients that received the influenza vaccine, 53% were treated with IFN-β-1a and 47% were not on DMT [99]. No differences were noted, where 93% of IFN-β-treated and 90.9% of non-treated MS patients achieved good post-vaccination hemagglutination inhibition titers [99]. During the 28-day period, no MS patient experienced a relapse [99]. A more recent and smaller IFN-β study corroborated the unhindered ability of these MS patients to mount a sufficient anti-influenza response [100]. An equivalent study design was deployed to examine the potential effect of teriflunomide on the response to trivalent influenza vaccine (TERIVA study) [101]. A high percentage of IFN-β-treated controls and both the 7 mg and 14 mg teriflunomide groups already had baseline high antibody titers towards H1N1 (88.4%, 82.5%, and 87.2%, respectively) and approximately half of the patients had high baseline protection towards the B strain (53.5%, 70%, and 69.2%%) and H3N2 (48.8%, 55%, and 53.8%, respectively) [101]. After the vaccination protocol, the 14 mg teriflunomide group resulted in a lower percentage of patients with sufficient H3N2 antibody titer when compared to the 7 mg teriflunomide and IFN-β control group (76.9% vs. 97.4% and 97.4%) [101]. That being said, all groups achieved the 70% proportion required by the European Criteria for Efficacy of Influenza Vaccination. In a randomized placebo-controlled trial, 138 MS patients were allocated to either 0.5 mg fingolimod or placebo groups (2:1 ratio) and six weeks later were vaccinated with the seasonal influenza vaccine [102]. Three weeks later, the responder rate for the influenza vaccine was 54% for the fingolimod-treated vs 84% for the placebo MS patients [102]. Furthermore, an even greater difference was seen at the 6-week follow-up (43% vs. 75% for fingolimod-treated and placebo MS groups, respectively) [102]. Although the differences between the groups were smaller and non-significant for an antigen-recall vaccination (tetanus toxoid booster dose), the fingolimod group did have a numerically lower proportion of responders [102]. A previous study with daclizumab, now withdrawn from the DMT market, also reported sufficient vaccinator response in an open-label single-arm extension study (termed SELECTED) [103]. The reported rate of seroconversion (from <10 hemagglutination inhibition titer to ≥40) was 69% for H1N1, 69% for H3N2, and 44% for B influenza strain [103]. Although the study had a sufficient ratio of responders as per EMA/FDA criteria, the lack of control arm limits the ability in deriving final conclusions. [103]. Other open-label, observational studies have utilized mixed patient populations which were treated with different DMTs and determined the overall influenza efficacy. For example, in a study of 102 MS patients, only 28.6% of natalizumab-treated cases achieved seroprotection towards H3N2, whereas the entire cohort had an average seroprotection rate of 72.5% [104]. Furthermore, the use of DMT was the only statistical predictor of not achieving seroprotection [104]. Interestingly, MS patients previously treated with other DMTs before their current therapy had significantly lower seroprotection when compared to those that were still on their first-choice medication (58% vs. 86% response rates) [104]. Lastly, the total cohort did not develop sufficient herd immunity of >40% seroprotected for influenza B strain [104]. During the 2009 influenza pandemic, a total of 113 MS patients and 215 controls were vaccinated with H1N1 swine flu vaccine [105]. When compared to the 43.5% response rate in controls, only MS patients treated with IFN-β achieved significant response (44.4% with HI titer of >40) [105]. On the other hand, a significantly lower response was seen in MS patients on glatiramer acetate (21.6%) or natalizumab (23.5%), and none of the MS patients on mitoxantrone developed protection towards influenza (0%) [105]. The same Author group provided a repeated study of a seasonal influenza vaccination where only MS patients on IFN-β, glatiramer acetate and non-treated MS patients showed comparable protection rates to the healthy controls [106]. On the other hand, both fingolimod and natalizumab-treated patients had significantly reduced protection rates [106]. Taken together, real-world observational studies do demonstrate lower rates of seroprotection when compared to the few aforementioned blinded vaccination trials.

The remaining studies utilize a range of vaccines that would test T-cell dependent recall response (tetanus-diphtheria toxoid vaccine), T-cell-independent humoral response (pneumococcal vaccine) and T-cell-dependent neoantigen response (meningococcal vaccine). An IFN-β-controlled study examined the effectiveness of vaccination in MS patients treated with dimethyl fumarate (DMF) [107]. There were no significant differences in T-cell-dependent recall response (both 68% of DMF-treated patients and 73% of IFN-β-treated patients had ≥2-fold increase in IgG), lower but non-significant T-cell independent humoral response (66% of DMF-treated vs. 79% IFN-β-treated) and no differences in the response towards neoantigen, with 53% of both DMF- and IFN-β-treated MS patients developing immunity towards meningococcal C [107]. A similar vaccine design was employed for a pilot study in 25 MS patients treated with alemtuzumab, which were compared to literature controls [108]. The T-cell dependent recall response was not assessed due to 100% pre-vaccination titer above the upper detection levels of the assay. On the other hand, the patients demonstrated a significant change from 13% to 91%, achieving seroprotection towards meningitis C [108]. Comparable to literature controls, 73% and 95% of alemtuzumab-treated patients demonstrated a 2-fold increase in antibodies towards pneumococcal vaccine serotypes 3 and 8, respectively [108]. It is important to note that only 2 out of 5 patients that received their vaccination within six months of alemutuzumab treatment did achieve a significant response [108].

Despite the fact that there are no MS studies that analyze vaccine response in settings of B-cell depleting therapies, such inferences can be derived from rheumatoid arthritis (RA) or NMO patients [109]. A study of 43 rituximab-treated RA patients showed diminished rates of achieving satisfactory levels of hemagglutination inhibition titer towards H3N2 (21% vs. 67% of matched controls) [110]. Furthermore, only 14% of rituximab-treated RA patients achieved immune response towards more than one antigen from the trivalent vaccine [110]. The seroprotection in these cases was not influenced by the additional methotrexate or prednisone treatment [110]. A similar lack of response has been shown in rituximab-treated lymphoma patients, where in comparison to 82% response rate in the control group, none of the 62 patients achieved sufficient vaccine efficacy [111]. Lastly, NMO patients receiving rituximab failed to mount sufficient titer of hemagglutination inhibition antibodies when compared to NMO patients treated with azathioprine or when compared to healthy controls [112]. However, these NMO patients did retain good T-cell-dependent recall response towards previously formed immunity [112]. Based on these studies, MS patients treated with B-cell depleting therapies may exhibit impaired vaccine response to influenza vaccination. A recent open-label, randomized Phase IIIb trial recruited 102 MS patients that were randomized either to ocrelizumab or no DMT group and assessed the humoral responses towards multiple vaccination protocols (VELOCE trial, NCT02545868). Patients treated with ocrelizumab had an attenuated humoral response when compared to the control group towards the tetanus vaccine (23.9% vs. 54.5%), pneumococcal vaccine (71.6% vs. 100%), and influenza vaccine (ranging from 55.6% to 80% in ocrelizumab-treated patients vs. 75% to 97% response in the control group, for 2015/2016 and 2016/2017 seasons, respectively).

### 3.3. Vaccines as Treatment for Multiple Sclerosis

Multiple attempts at developing MS-based vaccines have been previously explored and tested both in mice models and human subjects. Due to the vast number of studies, we will utilize a non-exhaustive list which will assist in demonstrating the variety of vaccine-targeted disease mechanisms. In the later 1990s, the initial increased interest in vaccination for MS led to multiple Phase I/II MS trials. One example showed that the commonly attempted T-cell vaccination principle does not deviate much from the typical vaccination paradigm. After retrieval and sorting of myelin-specific autoreactive T-cells from MS patient’s serum, the T-cells undergo in-vitro attenuation before being re-infused into the patients. The immune system therefore recognizes this attenuated agent (in this case myelin-specific T-cell) and mounts an immune response towards such circulating pathogenic T-cells.

Several small pilot studies preceded the patent acquisition and development of T-cell vaccination [113,114]. The first reported study utilized myelin-basic protein (MBP)-reactive T-cell lines derived from each subject, irradiated and re-infused back [113]. After the second inoculation, the patients demonstrated a significant decline in circulating MBP-reactive T-cells [113]. The study also suggested a potential induction of separate anti-clonotypic T-cells, which probably were able to recognize the variable region of the T-cell receptor and cause the depletion [113]. Two years later, a longitudinal study vaccinated five patients with RRMS and three with SPMS with autologous MBP-reactive and attenuated T-cells. The vaccinated patients demonstrated only an 8% increase in MRI-derived lesion volume, when compared to 39.5% in matched controls [114]. Another study utilized four SPMS patients which were inoculated every three months during a two year period with approximately 40 million bovine myelin-reactive and later attenuated T-cells per injection [115]. After the second cycle, a significant decrease in myelin-reactive and IFN-γ-producing T-cells was noted [115]. The cytotoxic effect was controlled by measuring the number of tetanus toxoid-reactive T-cells which remained stable [115]. However, there were no clinical effects, with two patients that remained stable, one that improved and one that worsened over the follow-up period [115]. After the aforementioned preliminary studies, an autoreactive attenuated T-cell vaccine named Tovaxin^®^ (later renamed as Tcelna^®^, Opexa Therapeutics) was tested in both Phase I/II and Phase IIb studies [116,117]. In the first smaller trial, isolated autoreactive T-cells reactive towards three major myelin antigens [myelin basic protein (MBP), proteolipid protein (PLP), and myelin oligodendrocyte glycoprotein (MOG)] were attenuated by γ-irradiation and administered back in 4 doses spanning over 20 week period [116]. Sixteen MS patients that had previously failed on other immunomodulatory medications were enrolled, and personalized vaccines were developed based on the peptide-specific T-cell reactivity [116]. The vaccination resulted in an early and significant 92.4% reduction in circulating myelin-specific T-cells, with a sustained 64.8% effect at week 52 [116]. The relapse rate analysis demonstrated a 85% decrease in the annual relapse rate; however, a significant part of the reduction could also be explained by regression to the mean. Disability analysis at the last follow-up visit at week 52 showed varied results [116]. Equal sets of 18.2% of MS patients showed either clinical improvement or worsening, respectively, whereas the remaining 63.6% remained stable [116]. The study showed only a mild to moderate adverse effect and no effect on the MRI-based contrast-enhancing lesions [116]. The second placebo-controlled, double-blind Phase IIb study was set to determine the efficacy of Tovaxin^®^ in Early Relapsing Multiple Sclerosis (TERMS study) and randomized 100 patients into treatment arm and 50 patients in placebo (2:1 ratio) [117]. A secondary post-hoc analysis included only 50 patients that had at least two or more relapses in the year preceding the study entry [117]. Analysis of the overall intent-to-treat group showed no significant clinical nor radiographic differences between the Tovaxin^®^-treated and placebo MS patients [117]. Within the MS group with annualized baseline relapse rate>1, Tovaxin^®^-treated patients had a trend of 56% reduction in relapses and a significant improvement in longitudinal EDSS scores [117]. The fact that the recruited MS patients were not DMT-naïve may have substantially reduced our ability to derive final conclusions. A rebranded Tcelna^®^ (Imilecleucel-T) drug was later further investigated in an additional Phase II trial in 183 SPMS patients, but the study did not meet the predefined primary end-point of lowering the brain atrophy nor the secondary endpoint of reducing the disability progression (NCT01684761).

Early experimental studies have shown that encephalitogenic T-cells express specific T-cell receptor (TCR) sequences within the complementary determining regions 2 and 3 (CDR2/CDR3) that can be recognized and targeted [118]. As such, one of the first proof-of-concept double-blind, placebo-controlled trials utilized vaccination with peptides specific towards Vβ5.2 expressing T-cells [119]. A group of 23 progressive MS patients were randomized in either 12-month vaccine treatment (100 µg weekly for four weeks, followed by monthly single doses) or placebo [119]. Six out of 17 vaccinated patients demonstrated an induced T-cell response towards the Vβ5.2 TCR compared to none of the placebo group [119]. Comparable to the experimental data, the responders also showed significant and persistent decrease in MBP-specific T-cells [119]. On the other hand, the non-responders and placebo group demonstrated an increase in MBP-specific T-cells [119]. There was a significant correlation between the frequency of MBP-specific T-cells and clinical outcomes [119]. In particular, patients with an increase in MBP-specific T-cells had disability progression, whereas the vaccine responders improved or remained clinically stable [119].

A Phase I/II trial tested trivalent TCR CDR2 peptide vaccine (containing BV5S2, BV6S5 and BV13S1 sequences) in 37 MS patients [120]. Immunological response to the vaccine was achieved in 94% of the adjuvant-vaccinated group, 14% in an non-adjuvant group, and 16% in the adjuvant-only group [120]. Responders to the vaccine had a significant induction of TCR-directed T-cells and each peptide acted as an individual immunogen [120]. During the 24-week study period, there were no clinical nor radiological differences between the responders and non-responders to the vaccination [120]. This was followed by an open-label single-arm study which recruited 23 MS patients (14 RRMS and 13 PMS) and assessed the mechanisms of action through which the mixture of the aforementioned three TCR-based peptides exhibited their effect [121]. After the vaccination, the patients had a significant increase in TCR-reactive T-cells, an effect that by the end of the trial returned back to the initial starting levels [121]. The vaccination also produced increase in IL-10-secreating T-cells and upregulated expression of FoxP3 within the regulatory T-cells [121]. Lastly, the vaccination also produced an expanded immunoregulatory network that spanned beyond the three TCR peptides and suppressed reactivity towards other neuroantigens [121]. Due to the financial problems of the original developer, no further development of NeuroVax^®^ is planned. Recently, the patent for the TCR peptide vaccine has been acquired by another pharmaceutical company (Immune Response BioPharma) which already initiated a Phase II double-blind, placebo-controlled SPMS study (NCT02149706). In 2014 FDA has designated NeuroVax^®^ with Pediatric Orphan Designation and Fast Track Designation for SPMS; however, no progress has been recently reported.

In light of the success seen in allergology, MS studies have also tried to induce antigen-specific tolerance by administration of analog peptides to the T-cell receptor (also termed as peptide vaccination). The form and preparation of the peptide varied from full length MBP, PLP, or MOG sequences, utilizing modified liposome packaging, and use of antigen-processing-independent epitopes (apitopes).

A double-blind, placebo-controlled RRMS study utilized an adhesive patch containing mixture of three different myelin peptides (PLP139-151, MOG35-55 and MBP85-99) over a period of one year [122]. Ten patients were randomly allocated to a placebo, 14 patients were treated with skin patch containing 1 mg of each peptide and four received a patch containing 10 mg of each peptide [122]. The transdermal application of myelin peptides resulted with formation of local and regional myelin-specific antigen-presenting cells [122]. The immunized patients also exhibited a significant decrease of myelin-specific CD4+ T-cells, greater levels of IL-10, and decreased levels of IFN-γ and TGF-β [122]. These changes were corroborated by significant clinical and radiological findings [122]. Both groups of RRMS patients treated with the 1 mg or 10 mg skin patch had significantly lower annualized relapse rate when compared to the placebo [123]. Only 19% of the 1 mg-treated RRMS patients had worsening in the EDSS scores when compared to 70% of the placebo [123]. Over the 12-month study period, patients in the 1 mg group showed a 19.7% decrease in T2 lesion volume and a 14.1% decrease in T2 lesion volume, whereas the placebo group had an increase of 25.4% and 61.2% in T2 and T1 lesion volumes, respectively [123]. Moreover, the 1 mg group also had a 66.5% reduction in the cumulative number of contrast-enhancing lesion [123]. Apart from mild skin reactions, no adverse events were noted [123]. The potential immunomodulatory efficacy, lower risk profile, and ease of administration makes such intervention an attractive treatment option. When compared to the current MS disease modifying treatments, the desensitization process towards myelin peptides does not significantly impact the entirety of the immune system and does not result in states of immune vulnerability.

In order to improve the desensitization, MBP peptides can be packaged in mannosylated liposomes which enhance their ability to bind with the dendritic CD206 receptor [124]. As such, the antigen-presenting cells can more efficiently activate regulatory T-cells and promote an anti-inflammatory effect [124]. A second synthetically produced mixture of four short MBP apitopes into one peptide vaccine named ATX-MS-1467 has been tested [125]. The preclinical MS mice models showed a significant dose-dependent effect in decreasing the mean overall disability, the peak disability scores and shortened the disease duration [125]. In a first Phase I trial, six SPMS patients received escalating doses up to 800 mg of ATX-MS-1467 and tolerated the medication relatively well [125]. No significant clinical effects were seen with a decrease in T-cell proliferative response towards MBP and a trend towards higher IL-10 gene expression [125]. Two additional Phase Ib and Phase IIa trials were recently conducted [126]. The first study had a primary outcome to determine the best mode of administration (intradermal vs. subcutaneous injections) and a secondary outcome of assessing the effect on MRI-detected contrast-enhancing lesions [126]. Based on the Phase Ib results, a faster intradermal 4-week titration period starting from 50 µg on day 1, 200 µg on day 15, and 800 µg on day 29 was utilized. This was followed by biweekly 800 µg injections for 16 weeks and an additional 16 weeks extension follow-up period [126]. Phase Ib study showed a significant 73% decrease in contrast-enhancing lesions (during 0−16 weeks) with a return to baseline values by the end of the observational period [126]. In comparison, the Phase IIa study showed a lower but significant decrease in contrast-enhancing lesions, an effect that persisted throughout the end of the observational period [126]. There were no changes in the predetermined disability outcomes with only a significant post-hoc finding of cognitive improvement [126]. Apitope^®^ has already announced plans for a Phase IIb, placebo-controlled, ATX-MS-1467 trial.

Other T-cell targets have been also tested. For example, findings on an MS-specific vaccine which attempts at targeting the voltage-gated potassium channel (Kv1.3) have been recently published [127]. These channels regulate the cell membrane potential, control the calcium-induced signaling and are highly expressed on activated T-cells, microglia, and macrophages [128]. Furthermore, pharmacological inhibition of Kv1.3 leads to lower pro-inflammatory cytokine production and diminished proliferation of autoantigen-specific T-cells [129]. In response, a prototype vaccine carrying both Kv1.3-specific epitope and promiscuous foreign T-cell PADRE epitope was created and termed PADRE-Kv1.3 [127]. After receiving two booster doses, the vaccinated rats exhibited stable and high levels of anti-Kv1.3 antibodies levels, with no side effects being observed [127]. When compared to placebo experimental autoimmune encephalomyelitis (EAE) groups, vaccinated EAE rats showed a significantly prolonged disease onset, lower initial disability peak and overall clinical disability scores [127]. The clinical features were accompanied by a lower frequency of pro-inflammatory T-cells (IFN-γ-producing and Th17), an increase in anti-inflammatory IL-10-secreting T-cells, lower T-cell infiltration, and a greater microglia shift towards an M2 subtype [127]. Experimental vaccination studies also attempt at upregulating the anti-inflammatory Treg cell populations [130]. A vaccine containing pVAX vector, which contained a *Mycobacterium leprae*-derived gene that encoded heat-shock protein 65 (hsp65), was produced [130]. After four doses of the pVAXhsp65 vaccine, the vaccinated EAE mice demonstrated higher levels of IFN-γ and anti-inflammatory IL-10 cytokines [130]. Despite the cytokine alterations and some clinical improvement, no peripheral nor CNS changes in the frequency of autoimmune CD4^+^/CD25^+^/Foxp3^+^ T-cells were noted [130].

An alternative vaccination strategy includes injection of DNA plasmids that contain selected antigens. The balance between an inflammatory immune response and induction of tolerance can be therefore controlled by the route, dose, and the modifications of the injected DNA-encoded antigen [131]. In the case of MS, DNA-based vaccines aimed to induce a greater rate of tolerance towards the myelin-based antigens and produce a shift towards a Th2 response [131,132].

Two MS DNA vaccine trials (both by Bayhill Therapeutics) have been conducted [133,134]. The first placebo-controlled study utilized a DNA vaccine containing full-length of the human MBP named BHT-3009 and recruited 30 RRMS or SPMS patients which were allocated to placebo, BHT-3009 alone, or BHT-3009 and atorvastatin [133]. No severe adverse events were reported and the study reported a non-significant trend of lower MRI-detected lesion activity [133]. Flow cytometric analysis showed that 5 out of 6 patients treated with the BHT-3009 showed a significant decline in IFN-γ-producing CD4+ T-cells, with the response lasting up to 50 weeks after initiation [133]. Furthermore, three patients underwent lumbar puncture both at baseline and follow-up visits and showed a decrease in CSF-derived autoantibodies towards not only MBP, but also towards PLP, MOG, and αB-crysallin [133]. This phase I/II trial demonstrated that DNA vaccination with myelin antigen did not produce immune activation, but rather produced tolerance towards multiple epitopes, a phenomenon called “bystander suppression”. A second larger phase II trial utilizing the same DNA vaccine was also conducted [134]. A final sample of 267 RRMS patients were randomized to placebo, 0.5 mg, or 1.5 mg monthly dose of BHT-3009 and were followed for 44 weeks [134]. The intervention did not result in any differences in terms of time to first relapse, annualized rate of relapses, and disability progression [134]. The lower dose of 0.5 mg BHT-3009 did have favorable but non-significant MRI outcomes, whereas the higher 1.5 mg dose did not have any effect [134]. During the 28-48 week period, the 0.5 mg BHT-3009 group had a trending 50% reduction in new enhancing lesions and at week 48 had a significant decrease in a mean volume of enhancing lesions when compared to the placebo [134]. The subgroup with CSF analysis showed that 0.5 mg of BHT-3009 produced a significant decrease in reactivity towards 23 different myelin autoantigens. In contrast, the 1.5 mg BHT-3009 group showed an increase in titers towards the examined PLP epitopes [134]. The authors have outlined the deleterious effect of the higher DNA vaccine dose to a greater percentage of immunostimulatory CpG motifs present in the DNA plasmid of the vaccine [134]. Due to funding restrictions, future investigation into this type of DNA vaccine remains uncertain. A recent joint effort between Genentech and Bayhill Therapeutics in development of similar type I diabetes vaccine may potentially revitalize the DNA vaccine MS program.

Lastly, MS-based vaccines targeting the B-cell pathway have been constructed. B-cell-activating factor (BAFF), alternatively called a B-lymphocyte stimulator (BLyS), is a potent modulator of B-cell differentiation and has been previously targeted in MS patients [135,136]. Therefore, by merging BLyS with tetanus toxoid T-cell epitope, the immune system will elicit an autoimmune response which will inhibit the in-vivo activity of BLyS [137]. BLyS-treated EAE mice created stable and high level of antibodies that caused a decline in B-cell proliferation and a decrease in clinical scores when compared to the control mice group [137]. Given that BAFF manipulation has historically resulted in negative MS disease outcomes, there is a need for better understanding the B-cell pathway before deploying such vaccines. A summary of the MS-based targets and the current stage of vaccine development is highlighted in Table 2.

## 4. Immunization Recommendations for Multiple Sclerosis Patients

Since the last published AAN practice guidelines regarding immunization in MS dating from 2002, epidemiological changes and emergence of new treatment protocols have drastically changed the overall disease settings [138]. Furthermore, interim reports have suggested that some immunization protocols may contribute to greater MS susceptibility and an increase in MS disease activity. Lastly, the significant immunosuppressive DMT effect may modulate the effectiveness of the vaccines by lowering the capability of the immune system to mount an effective response. In response to these new considerations, a new systematic review performed by the AAN has updated their practice guidelines [139]. Although some aspects were previously mentioned in this review, hereafter we will discuss the rationale and conclusion derived from the updated recommendations.

Eight recommendation statements encompassed aspects ranging from general immunization policies for MS patients, immunization during immunosuppressive and immunomodulatory drug use, and immunization during an active MS relapse. Firstly and most importantly, the AAN expert panel did not find definite evidence which would suggests that immunization can increase the risk for developing MS. Furthermore, they concluded that HPV, tetanus toxoid, pertussis, and smallpox vaccine may actually lower the risk for future MS diagnosis. That being said, vaccination of MS patients will not only contribute towards greater protection from developing infections, but also the MS population will continue to contribute to the herd immunity of the local community. When it comes to BCG vaccination, the AAN panel recommends following the WHO guidelines which vary based on the local prevalence of tuberculosis, children and older adults with negative tuberculin, or IFN-γ release assay and/or work-related risks involving health care and research, travel, and prison settings. The forth recommendation statement addressed the yearly influenza vaccination in MS patients on immunosuppressive therapy. As previously mentioned, some studies have suggested that there is an increased risk of MS exacerbation after influenza vaccination. The panel finds insufficient evidence to support or refute such associations. However, the recommendation does outline that patients treated with fingolimod, glatiramer acetate, mitoxantrone, and rituximab have lower responsiveness to the influenza vaccine. The panel agrees and recommends information provision by each individual prescribing these drugs. (Table 3) Infection screening guidelines before the initiation of such DMTs are also shown in Table 3. Apart from the prescribing information, the panel also provides a general encompassing statement which recommends against routine use of live-attenuated vaccines in patients that use or recently discontinued using DMT. Lastly, the guidelines suggest restraining of vaccination with live-attenuated vaccines in patients that are actively experiencing an MS relapse and during the period of three months after receiving high-dose systemic steroids. In line with the described AAN recommendations, a similar French-based recommendation has been issued. [140]. In addition to the MS-specific circumstances, the French providers responsible for the treatment and care of MS patients are recommended to follow the vaccination standards of the French High Council of Healthy which incorporates specific sections for immunocompromised populations [140].

Such recommendations provided by the major professional associations can contribute to more uniform vaccine use in settings of a chronic disease such as MS. The majority of the MS-based recommendations are similar in context when compared to other diseases like the autoimmune inflammatory rheumatic diseases (AIIRD). The European League against Rheumatism (EULAR) has issued comparable recommendations statements, including an inclusion of regular influenza vaccination, avoidance of live-attenuated vaccines, vaccination before the initiation of B-cell depleting therapy, and during stable disease periods. [141]. However, criteria formation through the use of only case-controlled studies as part of the meta-analysis process does have some limitations. The use of large international registries may help in identifying special circumstances and specific populations that exhibit alternative risk to benefit ratios.

## 5. Conclusions

Standard immunization protocols are not contributing towards greater MS risk and higher disease activity. In contrast to the aforementioned concerns, immunization strategies targeting certain pathogens may potentially provide additional clinical benefits that are pertinent to the overall health of the MS population. Despite the lower vaccination response rate, MS patients should be active participants in maintaining the overall herd immunity towards common vaccine-preventable diseases. Multiple vaccine/gene-based therapies continue to be developed and investigated in clinical trials; however, results to date have failed to achieve desired clinical outcomes. Treatment prospects including antigen desensitization and DNA vaccination are currently re-emerging as potentially viable new methods of targeting the complex MS pathophysiology. Before investing in costly research and development of new vaccine-based therapies, additional efforts targeting our emerging knowledge on disease mechanisms remains a priority.

## Figures and Tables

**Table 1 vaccines-08-00050-t001:** EBV-based molecular targets for development of EBV vaccine.

EBV Phase	Antigen/Gene	Function
Early - pre-latency	Zta	Transcriptional regulator
	Rta	Activator of the switch from latency to lytic cycle in B-cells
	EBNA2	Activation and stimulation of B-cell proliferation
	EBHRF1	Prevents host cell apoptosis and acts as Bcl-2 homolog
	EBNA-LP	EBNA2 co-activator and upregulated LMP1
Latency phases	LMP1	Acts as homologus CD40 and active TNF receptor
	LMP2A or 2B	Transmembrane proteins that mimic activated B-cell receptor
	EBNA1	Impairs MHC-I antigen presentation
Lytic	Divided in immediate, early and late lytic proteins	Working towards the goal of EBV-positive cell proliferation
EBV glycoproteins	gp350	Viral attachment to CD21 or CD35 of B-cells
	gB	Fusion protein
	gp42	Regulators of fusion
	gHgL	

EBV – Epstein-Barr virus, Zta – BamHI Z Epstein-Barr replication activator coded by BZLF1 gene, Rta – R transactivator coded by BRLF1 gene, EBNA2 – Epstein-Barr nuclear antigen 2, which upregulates viral expression of latent genes (LMP1 and LMP2A/B) and host cell gene expression (coding CD21, and MYC), BHRF1 – apoptosis regulator BHRF1 coded by the homonymous gene, which prevents host cell death, EBNA-LP – Epstein-Barr nuclear antigen leader protein, LMP1 – latent membrane protein 1 coded by the homonymous gene, which activates immune response signaling pathways (NF-κ-B family) and prevents apopotosis, TNF – tumor necrosis factor, LMP2A or 2B – latent membrane proteins 2A or 2B acting as activated B-cell receptor, gp – glycoprotein.

**Table 2 vaccines-08-00050-t002:** Vaccine development for multiple sclerosis treatment.

Vaccine	Name	Mechanism of Action	Latest Stage of Development	Clinical Trial Results
T-cell vaccine	Imilecleucel-T (Tcelna®)	Inducing immune response towards MS-specific T-cells	Phase II (*n* = 183 SPMS) and Phase II (*n* = 150 RRMS, randomized to T-cell vaccine or placebo)	- No effect in overall ITT group - Non-significant 56%- Significant reduction in relapses- No effect in SPMS
TCR vaccine	TCR-CDR2 peptide vaccine (NeuroVax®)	Targeting MS-specific T-cell receptor sequences	Phase I/II (*n* = 37 MS, randomized to vaccine or placebo)	- No clinical nor radiological effect
Peptide vaccine	MBP85-99, PLP139-151 and MOG35-55 mixture	Inducing tolerance towards MBP, LPL and MOG	Phase IIa (*n* = 30 RRMS, randomized to placebo, 1mg and 10mg mixture of antigens)	- Statistically significant 66.5% reduction in Gd-enhancing lesions compared to placebo
	ATX-MS-1467	Inducing MBP tolerance	Phase IIa (*n* = 37 RRMS, all on ATX-MS-1467)	- Statistically significant 46% reduction in Gd- enhancing lesions compared to pre-treatment period
DNA vaccine	BHT-3009	Inducing MBP tolerance	Phase IIb (*n* = 257 RRMS, randomized to placebo, 0.5mg and 1.5mg BHT-3009)	- Non-significant 50% reduction in Gd-enhancing lesions compared to placebo
B-cell pathway	TT-BLyS	Inhibit B-cell proliferation through blocking B-cell activating factor	Pre-clinical	-
Other targets	PADRE-Kv1.3	Inhibit voltage-gated potassium channel (Kv1.3)	Pre-clinical	-
	pVAXhsp65	Promoting regulatory T-cell proliferation through upregulation of HSP65	Pre-clinical	-

MS – multiple sclerosis, RRMS – relapsing-remitting multiple sclerosis, SPMS – secondary-progressive multiple sclerosis, CDR – complementarity-determining region, TCR – T-cell receptor, MBP – myelin basic protein, PLP – myelin protolipid protein, MOG – myelin oligodendrocyte glycoprotein, Gd – gadolinium, PADRE – pan HLD DR-binding epitope, Kv1.3 – potassium voltage-gated channel (shaker-related subfamily, member 3), TT-BlyS – tetanus toxoid – b-lymphocyte stimulator, HSP65- heat-sensitive protein 65.

**Table 3 vaccines-08-00050-t003:** Screening and live-vaccination before, during, and after immunosuppressive treatment of MS patients.

Prescribing Information	Screening	Vaccination before DMT	Vaccination during DMT Use	Vaccination after Discontinuation
Fingolimod	Active TB and Hepatitis B	1 month	Avoid live-vaccines	2 months
Teriflunomide	TB	-	Avoid live-vaccines	6 months
Alemtuzumab	TB^+^, varicella, VZV	6 weeks	Avoid live-vaccines	“recent” treatment
Ocrelizumab	Active Hepatitis B	4 weeks*	Avoid live-vaccines	until repletion

DMT – disease modifying treatment, TB - tuberculosis. * - 2 weeks for non-live vaccines as well. ^+^ - Despite the tuberculosis-positive MS were excluded from the alemtuzumab trials, the panel recommended this screening procedure. Furthermore, the panel adds hepatitis B and C to alemtuzumab screening, despite not being included in the prescribing information.

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
