# Peer review of "Infections, Vaccines and Autoimmunity: A Multiple Sclerosis Perspective"

_vaccines, 2020, doi:10.3390/vaccines8010050_

Round 1

Reviewer 1 Report

The current review by Jakimovski et al. summarizes the current status and advances in vaccination strategies in multiple sclerosis, particularly focusing on vaccination strategies targeting viruses. The work comprises good logical literary composition, and provides an informative summary of bacterial infection (with emphasis on H. pylori/C. pneumoniae, which may have opposing correlations with MS onset), viral infection and vaccination targeting viral antigens, and effects of immunization targeting non-viral components in MS. The review of the field appears to be quite thorough, and the recommendation summary from AAN is an excellent pre-concluding statement with respect to immunization strategies and MS. The review embodies detailed content which showcases the complex etiological association of MS with infection and immune response. Thus, the concerns here are editorial considerations that may improve clarity and focus of an otherwise, timely and informative review article.

1) The conclusion of this article seems to emphasize that no evidence is available to demonstrate that immunization is deleterious to MS onset. I suggest strengthening the impact of the conclusion by revising the conclusion to emphasize that vaccination strategies have not shown to adversely aggravate MS so far, that some immunization strategies also may show promise clinically.

2) The scope of the current review article is wide, I suggest distilling and summarizing the results into two additional tables describing viral-associated immunization targets and their outcome, and “other” non-viral targets and their effects on MS. Classifying and summarizing these clinical effects will preclude readers from having to trawl through details in the main text, and would provide a clear depiction of which vaccinations may show promise.

3) A brief discussion and comparison of current MS treatment strategies in the introduction may be helpful in emphasizing the potential importance of vaccination strategies in MS. Importantly, there is currently no cure for MS, and treatment strategies are largely focused on disease-modifiers; a brief discussion of current MS treatment regiments (steroids, relapse modifiers – gilenya, Tecfidera for example) may be helpful. Advantages to repurposing/repositioning acquired immune response through vaccination could also be highlighted here.  

Author Response

The Answers to both Reviewers are attached and also shown here (for each respectively).

General comment:

We thank both Reviewers for the constructive review and their contribution towards improving the overall quality of the manuscript. In the interim period, we additionally found one more relevant study (VELOCE MS, NCT02545868) that is now included in the manuscript. Point-by-point answers to all comments are shown hereafter:

Reviewer  1:

The current review by Jakimovski et al. summarizes the current status and advances in vaccination strategies in multiple sclerosis, particularly focusing on vaccination strategies targeting viruses. The work comprises good logical literary composition, and provides an informative summary of bacterial infection (with emphasis on H. pylori/C. pneumoniae, which may have opposing correlations with MS onset), viral infection and vaccination targeting viral antigens, and effects of immunization targeting non-viral components in MS. The review of the field appears to be quite thorough, and the recommendation summary from AAN is an excellent pre-concluding statement with respect to immunization strategies and MS. The review embodies detailed content which showcases the complex etiological association of MS with infection and immune response. Thus, the concerns here are editorial considerations that may improve clarity and focus of an otherwise, timely and informative review article.

Answer: We thank the Reviewer for the constructive recommendations.

1) The conclusion of this article seems to emphasize that no evidence is available to demonstrate that immunization is deleterious to MS onset. I suggest strengthening the impact of the conclusion by revising the conclusion to emphasize that vaccination strategies have not shown to adversely aggravate MS so far, that some immunization strategies also may show promise clinically.

Answer: In addition to the lack of concerns, we added a sentence stating that immunization protocols may even provide certain benefit to the MS patients.

2) The scope of the current review article is wide, I suggest distilling and summarizing the results into two additional tables describing viral-associated immunization targets and their outcome, and “other” non-viral targets and their effects on MS. Classifying and summarizing these clinical effects will preclude readers from having to trawl through details in the main text, and would provide a clear depiction of which vaccinations may show promise.

Answer: As both Reviewers have suggested, we have added two tables that summarize the current state of vaccine development for treatment of MS and another that describes the EBV-based targets in developing EBV vaccine. (now Table 1 and 2).

3) A brief discussion and comparison of current MS treatment strategies in the introduction may be helpful in emphasizing the potential importance of vaccination strategies in MS. Importantly, there is currently no cure for MS, and treatment strategies are largely focused on disease-modifiers; a brief discussion of current MS treatment regiments (steroids, relapse modifiers – gilenya, Tecfidera for example) may be helpful. Advantages to repurposing/repositioning acquired immune response through vaccination could also be highlighted here.  

Answer: We agree with the Reviewer. We added a short discourse regarding the current state of the MS medications and the re-occurring mechanisms of action that target the immune system in general but not the MS-specific pathology.

Reviewer 2 Report

Jakimovski and colleagues reported on the role of viruses, bacteria and their specific vaccinations in multiple sclerosis (MS); they have also discussed most recent advances in vaccination policies and vaccine-based treatments for MS. The review is overall clear and well written. Article research has been carried our accurately. I have added some suggestions to the authors below.

Paragraph 2.1. I would not mention Alzheimer, Parkinson, stroke, etc. The review already covers a wide issues, and I would not make it even wider.

Paragraph 2.2. It would be good having a figure summarizing possible EBV-related mechanisms, which are otherwise hard to follow throughout the text.

Paragraph 3.1. Considering the rise in the anti-vax movement, I believe this paragraph is crucial, and, in particular, I would suggest: 1) when referring to 80, authors have to clearly mention that there is no increased risk in the long term (as such the increased risk in the short term is not significant anymore over time, suggesting these cases were only anticipated rather than caused); 2) I would not mention the European Court, which is not a scientific body and whose decisions might be driven by a number of not-scientific factors; 3) at the end of the HBV paragraph, I would refer to Sesitile et al. Hum Vaccin Immunother 2018 showing that there is no increased MS risk following HBV vaccination on meta regression (which is definitely stronger evidence than descriptive review); 4) at the end of when mentioning BCG in RIS, I would add reference to https://clinicaltrials.gov/ct2/show/NCT03888924 ; 5) if possible, I would prepare a table showing the different studies relating vaccination to MS, along with design, sample, main findings.

Paragraph 3.3. Again, it would be good having a table dividing clinical trials based on the hypothesized vaccine mechanism of action, and reporting on clinical trial phase, sample and findings.

Paragraph 4. Authors could consider also including other immunization guidelines (e.g., French in Lebrun et al. Mult Scler Relat Disord 2019), and comparing with guidelines for other immunocompromised individuals.

Paragraph 5. In “Standard immunization protocols most probably are not contributing towards greater MS risk and higher disease activity”, “most probably” should be removed.

Author Response

Please see the attachment as well.

Reviewer 2:

Jakimovski and colleagues reported on the role of viruses, bacteria and their specific vaccinations in multiple sclerosis (MS); they have also discussed most recent advances in vaccination policies and vaccine-based treatments for MS. The review is overall clear and well written. Article research has been carried our accurately. I have added some suggestions to the authors below.

Answer: We thank the Reviewer for the constructive comments.

Paragraph 2.1. I would not mention Alzheimer, Parkinson, stroke, etc. The review already covers a wide issues, and I would not make it even wider.

Answer: This has been removed as suggested.

Paragraph 2.2. It would be good having a figure summarizing possible EBV-related mechanisms, which are otherwise hard to follow throughout the text.

Answer: This has been added in the manuscript as a table. (now Table 1)

Paragraph 3.1. Considering the rise in the anti-vax movement, I believe this paragraph is crucial, and, in particular, I would suggest: 1) when referring to 80, authors have to clearly mention that there is no increased risk in the long term (as such the increased risk in the short term is not significant anymore over time, suggesting these cases were only anticipated rather than caused); 2) I would not mention the European Court, which is not a scientific body and whose decisions might be driven by a number of not-scientific factors; 3) at the end of the HBV paragraph, I would refer to Sesitile et al. Hum Vaccin Immunother 2018 showing that there is no increased MS risk following HBV vaccination on meta regression (which is definitely stronger evidence than descriptive review); 4) at the end of when mentioning BCG in RIS, I would add reference to https://clinicaltrials.gov/ct2/show/NCT03888924 ; 5) if possible, I would prepare a table showing the different studies relating vaccination to MS, along with design, sample, main findings.

Answer: We have addressed the points suggested by the Reviewer. 1)  We added that the relapse in the vaccinated RIS patients could have been anticipated by the known data of the natural history progression. 2) This has been deleted. 3) We added and commented on Sesitile et al. references as suggested. 4) The NCT trial number was added. 5) Given that we conclude each paragraph for HBV and HPV with the reference to the latest meta-analysis, would additionally elongate the already extensive manuscript. We have included 2 full-page tables that describe the EBV targets and the vaccination trials in MS.

Paragraph 3.3. Again, it would be good having a table dividing clinical trials based on the hypothesized vaccine mechanism of action, and reporting on clinical trial phase, sample and findings.

Answer: This has been included in the manuscript as suggested. (now Table 2)

Paragraph 4. Authors could consider also including other immunization guidelines (e.g., French in Lebrun et al. Mult Scler Relat Disord 2019), and comparing with guidelines for other immunocompromised individuals.

Answer: In addition to the French immunization guidelines, we also included vaccination recommendations for other autoimmune chronic disease like the recently published European League Against Rheumatism (EULAR) in all autoimmune inflammatory rheumatic disease (AIIRD).

Furer, V., Rondaan, C., Heijstek, M.W., Agmon-Levin, N., van Assen, S., Bijl, M., Breedveld, F.C., D'Amelio, R., Dougados, M., Kapetanovic, M.C., van Laar, J.M., de Thurah, A., Landewe, R.B., Molto, A., Muller-Ladner, U., Schreiber, K., Smolar, L., Walker, J., Warnatz, K., Wulffraat, N.M., Elkayam, O., 2020. 2019 update of EULAR recommendations for vaccination in adult patients with autoimmune inflammatory rheumatic diseases. Ann Rheum Dis 79(1), 39-52.

Paragraph 5. In “Standard immunization protocols most probably are not contributing towards greater MS risk and higher disease activity”, “most probably” should be removed.

Answer: This is corrected as suggested.

Round 2

Reviewer 2 Report

Authors have addressed my concerns.